# Effect of the Similarity of Formulations and Excipients of Approved Generic Drug Products on In Vivo Bioequivalence for Putative Biopharmaceutics Classification System Class III Drugs

**DOI:** 10.3390/pharmaceutics15092366

**Published:** 2023-09-21

**Authors:** Ping Ren, Theresa Chan, Wen-Cheng Yang, Mitchell Frost, Yan Wang, Markham Luke, Myong-Jin Kim, Robert Lionberger, Yi Zhang

**Affiliations:** Division of Therapeutic Performance I, Division of Therapeutic Performance II, Immediate Office, Office of Research and Standards, Office of Generic Drugs, Center for Drug Evaluation and Research, U.S. Food and Drug Administration, 10903 New Hampshire Ave., Silver Spring, MD 20993, USA; ping.ren@fda.hhs.gov (P.R.); theresa.chan@fda.hhs.gov (T.C.); wencheng.yang@fda.hhs.gov (W.-C.Y.); mitchell.frost@fda.hhs.gov (M.F.); yan.wang3@fda.hhs.gov (Y.W.); markham.luke@fda.hhs.gov (M.L.); myongjin.kim@fda.hhs.gov (M.-J.K.); robert.lionberger@fda.hhs.gov (R.L.)

**Keywords:** Biopharmaceutics Classification System Class III, qualitatively (Q1)/quantitatively (Q2), formulation, excipients, biowaiver, pharmacokinetic (PK), bioequivalence (BE)

## Abstract

One of the potential essential factors that restricts generic industry from applying the Biopharmaceutics Classification System (BCS) Class III biowaiver is adherence to the stringent formulation criteria for formulation qualitative (Q1) sameness and quantitative (Q2) similarity. The present study has investigated formulations and excipients from 16 putative BCS Class III drug substances in a total of 19 drug products via 133 approved abbreviated new drug applications (ANDAs) containing in vivo bioequivalence (BE) studies in human subjects during the time period from 2006 to 2022. We included the BCS Class III drugs in this study by referring to published literature, the World Health Organization (WHO) BCS Class I-IV list, FDA internal assessments, and physicochemical properties (high solubility and low permeability) of specific drug substances. Based upon all 133 approved generic formulations in this study, the highest amount of each different compendial excipient with a total of 40 is defined as its corresponding typical amount that has not shown any potential impact on in vivo drug absorption. In the present study, although only 30.08% of the investigated generic formulations met Q1 the same/Q2 similar formulation criteria for the BCS Class III biowaiver, and while approximately 69.92% failed to meet those criteria with non-Q1/Q2 similar formulations, all test/reference ratios (T/R) and 90% confidence intervals for all instrumental PK parameters (AUC_0-t_, AUC_0-inf_, and Cmax) met the bioequivalence (BE) criteria (80–125%). The results of formulation assessment suggest that the commonly used excipients without atypical amounts did not impact absorption of 16 putative BCS Class III drug substances. The rate and extent of absorption of drugs appears to be more dependent upon the biopharmaceutic and physiochemical properties of BCS Class III drug substance and less, or not dependent upon their formulations, excipients, and the excipients class. Our findings may lead to a more flexible formulation design space regarding the stringent BCS Class III formulation criteria.

## 1. Introduction

The Biopharmaceutics Classification System (BCS), based on aqueous solubility and intestinal permeability property, has classified drug substances into four classes (identified as BCS Classes I-IV). These classes have been widely used to predict drug absorption during pharmaceutical development. More recently, the US Food and Drug Administration (FDA) has employed BCS-based biowaivers in regulatory practice and published a final guidance titled “*M9 Biopharmaceutics Classification System-Based Biowaiver*” (referred to as “M9 BCS-Based Waiver Guidance for Industry” in this article) [1]. This recent guidance describes the BCS biowaiver approach, by which the firm may submit a waiver of in vivo bioavailability (BA) or BE studies for an immediate-release (IR) solid oral dosage form if the drug substance has high solubility with either high (BCS Class I) or low permeability (BCS classes III). 

According to the M9 BCS-Based Biowaiver Guidance for Industry, highly soluble drug substances with low permeability when the absolute bioavailability is <85% are classified as BCS Class III drug substances. Due to their low permeability characteristics, they are more susceptible to the effect of excipients, which may have site-specific absorption impacts [2]. As a result, in order to be eligible for the waiver based upon BCS Class III drugs, the drug products should meet additional requirements regarding dissolution and formulation when compared to the criteria for biowaiver based upon BCS Class I drugs. Besides the criterion for very rapid in vitro dissolution rate, generic formulation should be Q1 the same and Q2 similar to the reference listed drug (RLD) in order to be eligible for the BCS-based Class III biowaiver route. Q1 sameness is defined as the generic product using the same inactive ingredient(s) as the reference product. Q2 similarity is defined as having within ± 10% of the amount of excipient in the reference product, and the cumulative difference for these excipients should be within ± 10%. Within the context of quantitative similarity, differences in excipients for drug products containing BCS Class III drugs should not exceed the individual criterion of specific excipient class (i.e., filler, disintegrant, etc.) [1]. These restrictions are driven by the concern that excipients may have a potential impact on the absorption of drugs with low permeability in vivo. For instance, previous studies have found that excessive quantities of certain unusual excipients, such as sugar alcohols (sorbitol and mannitol) and polyethylene glycol (PEG) 400, can influence intestinal permeability by altering transporter mechanisms and modifying gastrointestinal (GI) transit time. These unusual excipients were reported to affect the rate or extent of absorption of BCS Class III drug substances, such as ranitidine or cimetidine [2]. 

Recently, the effect of commonly used excipients on the oral absorption of BCS Class III drugs was also evaluated. The permeability of four BCS Class III substances (acyclovir, atenolol, ganciclovir, and nadolol) was assessed with five commonly used excipients using an in vitro Caco-2 cell monolayer system and an in-situ rat intestinal perfusion model [3]. No substantial increases in the permeability of these drug substances were observed in the presence of the tested excipients in either of the models. Moreover, the in vivo BE studies assessed the impact of very large amounts of 14 commonly used excipients on BCS Class III drugs’ absorption using cimetidine and acyclovir as model substances in humans [4]. The results of these in vivo BE studies demonstrated that all commonly used excipients had no significant impact on bioavailability of BCSIII drug substances, except hydroxypropyl methylcellulose (HPMC) and sorbitol. Both excipients exhibited lower absorption of cimetidine and acyclovir. Since these studies only investigated two drugs, it is possible that other BCS Class III drugs have properties that differ from cimetidine and acyclovir, rendering those drugs susceptible to other excipient influences that may cause modified drug absorption [5]. To date, very limited in vivo human studies have explored commonly used excipients that may alter the bioavailability of BCS Class III drugs.

Since the biowaivers have been extended to BCS Class III drugs by the recently revised publication of the M9 BCS-Based Biowaiver Guidance for Industry [1], generic industry has shown significant interest in applying BCS Class III biowaiver approach to ANDAs as well as seeking out clarifications on meeting formulation similarity criteria and effects of various excipients on in vivo BE performance for BCS Class III drugs via submitting controlled correspondences, pre-ANDA drug development meeting requests/pre-submission meeting requests, or BE comments to dockets to the Agency. The generic industry has been asking questions regarding the list of drug substances eligible for BCS Class III biowaiver, formulation similarity assessment, solubility data analysis, permeability requirement, narrow therapeutic index identification, in vitro approaches for biowaiver, different excipient classes, a single active ingredient BCS Class III biowaiver in fixed dose combination drugs, and dissolution profile comparison, drug degradation, etc. The Office of Generic Drugs (OGD) has invested substantial amount of time and effort in assessing and responding to those inquiries and conducted extensive research on the generic industry’s interests and questions on BCS Class III biowaiver. 

BCS Class III drugs constitute roughly 25% of drugs marketed in the United States. Moreover, nearly 40% of orally administered drugs on the World Health Organization (WHO) Model List of Essential Medicines are BCS Class III drugs [6]. Generally, generic industry may submit an ANDA containing in vivo BE studies with PK endpoints or comparative clinical endpoints BE study (CCEB) to FDA to obtain generic drug approvals; however, it may take a significant amount of time for the application to be assessed and approved prior to the drugs entering the market. Furthermore, in vivo BE studies with PK endpoints or CCEB approaches may need to recruit a large number of human subjects, which could significantly increase the financial investment burden and unnecessary human exposure. The BCS Class III-based biowaiver can be a beneficial alternative BE approach that may reduce development costs, decrease unnecessary human exposure to drugs, and expedite the drug approval process. Meanwhile, it may be critical in ensuring high quality, effective, and affordable medicines available to the American general public. Lastly, extending biowaivers to BCS Class III drugs, particularly for drug products only targeting a specific small population (e.g., drugs with orphan designation), drug products in shortage, drug products with commercially unavailable/discontinued reference drug product (RLD/RS), and drug products having difficulties in recruiting patients for in vivo BE studies especially for oncology medications, could potentially promote the development and approval of generic drugs, which will ultimately improve patients’ access to more affordable medicines in their time of need. 

The present study was conducted to assess the impact of Q1/Q2 formulation similarity and excipients on the absorption in human for 16 putative BCS Class III drugs that were retrospectively collected from previously approved ANDA submissions. Our research results have provided further scientific evidence to support potential BCS-based waiver of in vivo BE studies for Class III drugs. This work allowed us to reveal the potential effect of biopharmaceutical properties of drug substances, formulation of drug products, and individual excipients on the in vivo performance of BCS Class III drug products. Furthermore, our findings in the present study may help to explore more flexibility for biowaiver recommendation regarding the conservative formulation criteria of Q1 sameness/Q2 similarity for BCS Class III drug products.

## 2. Materials and Methods

Based upon the biopharmaceutic and physiochemical properties, we identified a total of 16 putative BCS Class III candidate drug substances involving total of 19 IR drug products formulated as either tablets or capsules. Based on the current FDA Orange Book, 133 approved ANDAs were surveyed in the period between March 2006 to September 2022 for the 19 selected IR products. The retrospectively collected data from these ANDAs were analyzed regarding the high solubility and different permeability ranges delivered to the systemic circulation. The formulation compositions were compared and categorized based on Q1 sameness and Q2 similarity between the generic (test) and reference products in each case. The sameness, similarities, and differences of the generic drug formulations to the corresponding reference products were classified into four groups: Group of Q1/Q2 same formulation containing the same inactive ingredients with individual excipient difference within ±5%; Group of Q1 same/Q2 similar formulation containing within ± 10% of the amount of excipient in the reference product and the within ± 10% cumulative difference for these excipients; Group of Q1 same/Q2 different formulation containing a total additive effect of all excipient changes that is greater than 10%; Group of Q1 different formulation containing different excipient(s), either added or substituted. 

The excipients used in the studied BCS Class III drug formulations were analyzed by their functions, frequency of use, range of weight per dosage unit (mg), and percentile per total weight (*w*/*w*). The Q2 changes in different excipient classes, such as filler, binder, disintegrant, binder, lubricant, and glidant for individual formulation of generic drug products were also assessed based on the criteria set forth by M9 BCS-Based Biowaiver Guidance for Industry compared to the reference product.

In addition, all pivotal PK parameters, including area under the plasma concentration-time curve (AUC_0-t_, AUC_0-i_) and peak concentration (C_max_), were collected from a total of 217 in vivo BE studies with PK endpoints in the 133 approved ANDAs. All BE studies were single-dose, open-label, randomized, two-way or three-way crossover in vivo BE studies under fasting and fed conditions in healthy human subjects except for one approved ANDA, which contained a multi-dose, steady state BE study in patients. The BE studies consisted of 133 fasting and 84 fed BE studies. Box and whisker plots were constructed to exhibit the distribution of the Test/Reference (T/R) ratios and 90% confidence intervals (CIs) of the test/reference geometric mean ratios for C_max_ and AUC based on low or moderate permeability from all investigated approved ANDAs. 

## 3. Results

### 3.1. General Information

As per the published literature [7], WHO BCS Class III list [8], the FDA’s internal assessments, and the RLD labelings, 16 drug substances with systemic action in high solubility and low permeability were selected as putative BCS Class III drug substance candidates, as summarized in Table 1. The drugs that cannot be delivered to the systemic circulation with almost zero permeability, narrow therapeutic index drugs, and drugs with unstable solubility and permeability with >10% degradation were excluded from investigated list, according to the M9 BCS-Based Biowaiver Guidance for Industry. A total of 19 drug products (tablets: n = 13 and capsules: n = 6) were assessed in the present study, four of which have orphan designation status and six of which are on the drug shortage list, as well as one with BE recommendation for patient subjects with anti-oncology drug. All collected ANDAs were approved by either both in vivo fasting and fed BE studies or an in vivo fasting BE study only (such as penicillamine and trientine). Currently, the US FDA has approved six drug products via the BCS-based Class III biowaiver regulatory pathway; these are rasagiline mesylate tablets, oseltamivir phosphate tablets, hydroxychloroquine sulfate tablets, tipiracil hydrochloride; trifluridine tablets, trientine hydrochloride capsules, and migalastat hydrochloride capsules. 

### 3.2. Biopharmaceutic Properties

Solubility

According to the M9 BCS-Based Biowaiver Guidance for Industry, a drug substance is considered highly soluble if the highest single therapeutic dose is completely soluble in 250 mL or less of aqueous media across the physiological pH range of 1.2–6.8 at 37 ± 1 °C. The solubility data for each drug substance were collected from either ANDA or new drug application (NDA) submissions. The list of investigated drug substances, along with pH range, solubility value, the highest single therapeutic dose, and the highest strength, are presented in Table 2. The reported lowest solubility value over the pH range of 1.2–6.8 for all drug substances was higher than that of the highest single therapeutic dose except acyclovir. Acyclovir is the only one drug substance that failed to meet the highly soluble criteria for the highest single therapeutic dose. As per the RLD labeling, increases in plasma acyclovir concentrations were less than dose proportional over the dosing range (200 to 800 mg). Based upon the M9 BCS-Based Biowaiver Guidance for Industry, acyclovir may not be classified as a highly soluble drug by its solubility data and dose proportionality results. However, intra-Agency agreements and many published studies [9,10] classify this drug substance as either BCS Class III or IV [11]. Thus, 16 potential BCS Class III drug substances, including acyclovir, could be considered as highly soluble.

Permeability

For all 16 potential BCS Class III drug substances, the permeability ranged from 6% to 83%, as per corresponding RLD labels. Based on the permeability data, the drug substances were further divided into two sub-groups: one for low (fa < 50%) permeability and another for moderate (fa = 50–84%) permeability, as shown in Table 3. All permeability data were obtained from absolute bioavailability studies, and some were also supported by in vitro Caco-2 permeability assays and in vivo mass balance studies. The majority of drug substances were associated with the effects of efflux and different absorptive transporters on oral drug absorption, as listed in Table 3. Eight out of sixteen drug substances with limited permeability may be attributable to P-glycoprotein (P-gp)-mediated efflux transport. By referring to the data from internal and external sources, all 16 potential BCS Class III drug substance candidates selected were characterized with high solubility and low permeability (<85%), as defined by the M9 BCS-Based Biowaiver Guidance for Industry.

### 3.3. Formulation Assessment 

The component and composition of formulations for 19 putative BCS Class III drug products from 133 approved ANDA submissions were collected in the present study. The formulation assessment of the generic products from these ANDAs has shown that 6.02% (8/133) of formulations are Q1/Q2 same, 24.06% (32/133) are Q1 same/Q2 similar, 17.29% (23/133) are Q1 same/Q2 different, and 52.63% (68/133) are Q1/Q2 different when compared to their corresponding reference drug product formulations. Approximately 69.92% of the investigated generic drug formulations were non-Q1 same/Q2 similar to their corresponding reference drug products, in which they either exceeded ±10% of the amount of excipient or ±10% of the cumulative difference for variety of excipients in the reference product. Only 30.08% of the studied generic formulations met the BCS Class III quantitative similarity of the formulation biowaiver criteria set in the M9 BCS-Based Biowaiver Guidance for Industry and, thus, are eligible for the BCS-based Class III biowaiver as depicted in Figure 1. 

### 3.4. Excipient Analysis

General information

A total of 40 compendial excipients with various amounts, which are listed in the current FDA-approved immediate-release solid oral dosage forms, were used to formulate 133 generic drug products in approved ANDAs for 19 BCS Class III drug products candidates. Table 4 shows individual excipients that were observed in at least 5% of the ANDA formulations studied. Among the approved ANDAs, the top 10 commonly used excipients were magnesium stearate, microcrystalline cellulose, povidone, starch, lactose, colloidal silicon dioxide, sodium starch glycolate, croscarmellose sodium, pregelatinized starch, and stearic acid. Based upon the range of weight per unit, as illustrated in Table 4, there is no atypical amount used in these formulations (atypical amount is defined as over the upper limit of the Range of Weight per unit (mg/unit) from 133 formulations with acceptable in vivo BE studies).

Potential impacts of unusual excipients on absorption

Among the 133 ANDAs, a few contained some unusual excipients with known or suspected effects on drug absorption, such as sodium lauryl sulfate (SLS), mannitol, Polysorbate 80, and hydroxypropyl methylcellulose (HPMC or hypromellose). More than 13% of investigated generic formulations contain SLS, with the weight (mg) per dosage unit ranging from 0.3 mg to 6.3 mg and *w*/*w*% ranging from 0.1% to 1.5%. The sugar-alcohol excipients, mannitol and isomalt, were also observed in five formulations, with weight (mg) per dosage units from 44 to 111.85 mg, the range of *w*/*w*% from 6.29% to 48.53% and 114.24 mg with 95.2%, respectively. Approximately 5.34% generic product formulations contained HPMC. None of the drug products contained PEG 400 or sorbitol. There were also no atypically large amounts of unusual excipients in these ANDAs (Table 5).

Excipient assessment

In addition to assessing the amount of excipients that exceeded ± 10% (*w*/*w*) or within ±10% (*w*/*w*) of the cumulative difference for a variety of excipients, the Q2 difference in excipients are also assessed by excipient class according to the M9 BCS-Based Biowaiver Guidance for Industry. Each excipient class should not exceed allowable weight percentage (%*w*/*w*) of the core formulation, as described in Table 6. Deviations from those criteria may require appropriate justification for each ANDA submission. The breakdown of the excipient weight percentile changes (%) by excipient class and Q1/Q2 classification groups are illustrated in Figure 2. Overall, the results showed that lubricant (stearates) exhibited the highest frequency of exceeding the percent change (%*w*/*w*) in the observed excipients, followed by filler, disintegrant (starch and other), binder, and glidant (talc and other). Particularly, magnesium stearate, microcrystalline cellulose, povidone, lactose monohydrate, and sodium starch glycolate are ranked the top five excipients with the most frequency in excipient class changes. Among the four formulation groups classified by Q1/Q2, generic formulations classified as Q1/Q2 same did not have any observed changes that exceeded for any excipient class. Conversely, the largest number of Q2 changes in excipient class were distributed in the non Q1/Q2 similar formulation groups. For Q1 same/Q2 similar formulation group, only a few excipient changes were observed in binder, disintegrate-other, glidant-other, and lubricant-stearates.

### 3.5. In Vivo Bioequivalence Studies

The general information of 217 BE studies corresponding to the drug substance permeability is summarized in Table 7.

The distribution of PK parameters (AUC_0-t_, AUC_0-i_ and C_max_), of T/R ratios, and 90% CIs of the T/R geometric mean ratios were collected from a total of 217 BE studies, including 133 fasting and 84 fed BE studies, as illustrated in Figure 3. The statistically calculated 90% CI values for T/R ratios of AUC and C_max_ were within the acceptable limit of 80–125% for all ANDA submissions, except a few in the group of low permeability as illustrated in Figure 3. Due to the high variability in PK of those drug products, a few BE studies used three-way replicated design with the reference-scaled average BE approach. Although their 90% CI results of C_max_ barely failed to meet the 80–125% limit, their 95% upper confidence bound were ≤0, which meets the BE criteria. The study results demonstrated that the investigated generic drug products in all 217 BE studies for the 19 generic BCS III candidate drug products were bioequivalent to their corresponding reference products.

Of the 133 ANDAs, only 7 ANDAs were found to contain failed or inadequate fed and fasting studies in the application package. Of these failed or inadequate fed and fasting studies, four (57.14%) were due to statistical deficiencies with small sample size, one (14.29%) was due to bioanalytical deficiencies, and two (28.57%) were due to lack of information related to amount of ink printing. None of the failed BE studies were relevant to formulation differences. 

## 4. Discussion

In the present study, 16 putative BCS Class III drug substances were chosen as the model drugs to explore the impact of the similarity of formulations and excipients on in vivo absorption of those drugs in human from 133 approved ANDAs with different IR solid oral-dosage forms. Each drug absorption was assessed by AUC_0-t_, AUC_0-inf_, and C_max_ in the single dose in vivo BE studies with PK endpoints under the fasting and fed states or fasting only condition in healthy subjects. Only one approved ANDA was approved based upon multi-dose, two-period crossover in vivo steady-state BE study in cancer patients. A total of 40 compendial excipients in amounts typically used in IR solid oral dosage forms for 133 different formulations in these approved ANDAs were assessed. Our results of in vivo BE studies in the 133 different generic drug formulations have demonstrated the bioequivalence between the test and reference products under fasting and fed conditions in human subjects, regardless of if the generic formulation was Q1 same/Q2 similar or non Q1/Q2 similar to its corresponding reference product. Thus, different formulations with commonly used excipients in varied, but yet typical, amounts did not appear to affect the rate and extent of absorption of their respective putative BCS Class III drug products.

According to the M9 BCS-Based Biowaiver Guidance for Industry, only the Q1 the same/Q2 similar formulations are eligible for a biowaiver through in vitro approaches for BCS Class III drugs. We further divided the 133 different generic drug formulations into Q1 the same/Q2 similar and non-Q1/Q2 similar formulations. There are approximately 30.08% of generic drug formulations in our study that met the Q1 same/Q2 similar formulation definition set by the BCS Class III biowaiver criteria. The results of in vivo BE studies with those formulations demonstrated the BE between the test and reference products under fasting and fed conditions. Therefore, the in vivo BE outcomes strongly support the biowaiver of in vivo BE studies for Q1 same/Q2 similar generic formulations through in vitro approaches. It is reasonable to predict that the formulation with Q1 sameness/Q2 similarity is able to ensure that test products are bioequivalent to their reference products when their dissolution tests are very rapid across all physiologic pH conditions. On the other hand, in vivo fasting and fed BE studies also demonstrated that majority (69.92%) of the investigated ANDAs with non-Q1 same/Q2 similar generic formulations were bioequivalent to their corresponding reference products. These results suggest a similar conclusion as the previous in vivo BE studies on 12 common excipients, suggesting that a majority of commonly used excipients need not to be Q1 the same nor Q2 similar to the reference to establish BE between generic and reference drug products [4]. Our findings indicate that the excipients used in these non-Q1/Q2 similar formulations may not appear in modulating in vivo intestinal absorption in humans. 

When excipient differences exist, we should evaluate whether the excipient differences may affect the absorption profile of drug substance in order to determine if a BCS-based waiver is applicable. Thus, possible effects of an excipient on in vivo absorption should be assessed by considering the mechanism by which the excipient may affect absorption and amount of excipient used in the formulation. Among the 40 compendial excipients used in the 133 generic drug formulations collected in 19 putative BCS Class III drug products, a majority of those excipients did not have any effects on the drug solubility/dissolution, GI motility, transit time, gut wall metabolism, and intestinal permeability. In addition, none of them were used in an atypical amount in those formulations. Thus, those excipients may not pose any impact on intestinal absorption, such as microcrystalline cellulose, povidone, starch, lactose, colloidal silicon dioxide, sodium starch glycolate, croscarmellose sodium, pregelatinized starch, and talc. Our investigation on the in vivo BE studies demonstrated that 16 commonly used excipients (>5% of ANDAs) in the 133 generic drug formulations did not significantly affect in vivo absorption of these investigated BCS Class III drug products in humans. A similar conclusion has been reported by previously published studies [3,4], evaluating microcrystalline cellulose, povidone, starch, lactose, colloidal silicon dioxide, sodium starch glycolate, croscarmellose sodium, pregelatinized starch, stearic acid, and talc through Caco-2 cell monolayer system, in situ rat intestinal perfusion, and in vivo BE studies [12,13]. In summary, these excipients, with a typical amount used to formulate these generic drug products, do not appear to affect the rate and extent of absorption of the BCS Class III drug products. 

With few exceptions, magnesium stearate was the most commonly used excipient as a lubricant in more than 77.44% of the studied ANDAs distributed across all 19 putative BCS Class III drug products. Magnesium stearate usually forms a water repellent coat around individual granules, which may inhibit the wettability of solid dosage form drug products and decrease the penetrability of dissolution medium thereby reducing the effective surface area and prolonging the process of dissolution. This may, in turn, reduce bioavailability. The large amount of this excipient was reported to reduce dissolution release rate and PK parameters, such as AUC and C_max_, due to over lubrication with atypical large amount (40 mg/unit) [4]. Our finding in excipient analysis showed that the formulations containing magnesium stearate at the range of 0.5–20 mg/unit with a range (%*w*/*w*) of 0.25–3.33% across all investigated ANDAs, which is far less than 40 mg/unit. All in vivo BE studies showed that the generic drug products containing this excipient demonstrated the bioequivalent to their corresponding reference drug products. This result indicates the amount of magnesium stearate, such as less than or equal to 20 mg/unit, that may not influence adversely bioavailability of BCS Class III drugs in human. 

On the other hand, sodium lauryl sulfate (SLS), as a lubricant and surfactant, has the enhancing effect of dissolution release rate due to an increased wetting and better solvent penetration. It may also increase the drug’s capability to cross the liquid barriers of the cell membrane at the absorption site in order to enter the systemic circulation. The previously published studies have reported that SLS did influence the permeability of almost all drugs tested by reversibly opening tight junctions in Caco-2 cell culture [12] and enhancing permeability coefficients in jejunum and ileum from the in-situ intestinal perfusion studies [9,14]. Previous studies have demonstrated that SLS at concentrations of 0.139 mM [12] and 0.4 mM [15] can reversibly open tight junctions in Caco-2 cell culture. The in-situ intestinal perfusion studies found that the small intestinal permeability increased with increasing concentrations of 0.015% as well as 2%, 3%, and 4% [14]. SLS can change intestinal permeability with different mechanisms to have effect on the rate limiting step in the absorption of BCS Class III drugs. Approximately 13.74% of the generic drug formulations from 133 ANDAs used this excipient as a lubricant at much low concentrations when compared to the previously published results, approximately with a concentration range from 0.0003% to 0.008% with the highest daily single therapeutic dose. Our results, showing no influence on absorption of BCS Class III drug products, may be due to the low concentrations of SLS used in these formulations.

Less than 5% of the approved ANDAs were formulated with polysorbate 80, sugar alcohol, and osmotic agent in our present study. Since BCS Class III drugs are often associated with site-dependent absorption characteristics, those unusual excipients have been found to be capable of influencing drug absorption or bioavailability through a variety of mechanisms, such as modification in solubility/dissolution, change in intestinal permeability, and modulation of GI motility [16,17]. As a result, they are considered as unusual excipients that may significantly alter drug absorption [18]. The reason that all formulations with those excipients in the present study met in vivo BE criteria may be because the concentration was not sufficient to reach effect threshold of those excipients [19]. Also, it is possible that different BCS Class III drugs have different biopharmaceutic and physiologic properties to render varied susceptibility to those unusual excipient influences that cause modified drug absorption [20]. Due to limited approved ANDAs with a small amount of these unusual excipients, the exact influence on permeability by these excipients were inconclusive. In future, dose-response relationships on excipients with individual or different combinations need to be established in order to better understand these unusual excipients’ impact on in vivo absorption of BCS Class III drug products. Interestingly, unusual excipients are well-described in published studies, but very few generic drug products were formulated with those unusual excipients (except SLS) in a total of 133 ANDAs investigated in the present study. Moreover, none of the studied generic formulations in the present study contains PEG 400 or sorbitol. It seems unconvincing to use the information on these unusual excipients, such as PEG 400, sorbitol, and mannitol, to predict in vivo performance of generic BCS Class III drugs. 

In addition to the criteria of formulation Q2 similarity set by M9 BCS-Based Biowaiver Guidance for Industry, the other weight percentile change criteria with different excipient classes, such as filler, binder, or lubricant, etc. are also applied to assess formulation Q2 similarity. However, they are rarely mentioned in recently published studies. Our findings indicate that the large number of excessed weight percentile changes (%) of different excipient classes were mainly observed in the non Q1/Q2 similar formulation group. As per the in vivo BE studies in non-Q1 same/Q2 similar formulation group, our results demonstrated that the weight percentile changes (%) of different excipient classes did not appear to modulate the rate and extent of absorption of their respective BCS Class III drug products. In addition, within the context of Q1/Q2 similarity, the changes for individual excipient class in binder, disintegrate-other, glidant-other, and lubricant-stearates were observed in four Q1/Q2 similar formulations. As per the current M9 BCS-Based Biowaiver Guidance for Industry, a total of 10% (4/40) formulations in Q1/Q2 similar group did not meet the criteria set for individual excipient class, even though they met the total additive changes for formulation Q2 similarity. 

As discussed above, the results of our BE studies with different generic drug products in the IR dosage forms of tablet and capsule have provided a satisfactory assurance that BCS Class III drug products can be granted with a biowaiver using an in vitro approach instead of in vivo BE studies when the generic formulation is Q1 same/Q2 similar to the corresponding reference product. Moreover, our results showed that majority (69.92%) of the 133 ANDAs investigated employed non-Q1/Q2 similar formulations that are not eligible for BCS III waivers according to the current M9 BCS Class III biowaiver criteria. However, they are all bioequivalent to their corresponding reference products. Our study results could readily make a conclusion that 16 BCS Class III generic drugs formulated with variations in commonly used excipients without atypic amounts may not influence in vivo bioequivalence to the corresponding reference product, regardless Q1/Q2 similar or non Q1/Q2 similar formulations. This may be due to the critical physicochemical characteristics of high solubility for these BCS Class III drug substances. With very rapid dissolve, high aqueous solubility of BCS Class III drug substance may behave in vivo like an oral solution, for which, the in vivo bioavailability or BE of drug product may be self-evidence. The requirement for submission of evidence obtained in vivo BE studies should be waived. However, membrane permeability is expected to be the rate-limiting step in drug absorption of BCS Class III [13]. They are considered to be more susceptible to the effects of excipients on permeability compared to BCS Class I drug. These drugs may have site-specific absorption, so there are a greater number of mechanisms through which excipients can affect their absorption than for BCS Class I drugs [1]. However, our in vivo fasting or fed BE results demonstrated that the commonly used excipient without atypical amount may not influence the permeability of BCS Class III drug. No published scientific literature reported those excipients may modify physiological factors, such as gastric emptying and GI transit time as well as membrane permeability (except SLS with atypical amount). The non-Q1Q2 formulations with different excipient classes may not show different bioavailability of BCS Class III drugs. Our results indicated the requirements of Q1/Q2 similar formulation and no difference in excipient class may potentially be overly stringent causing a significant barrier to extend the application of BCS III-based biowaivers for an ANDA submission. Our study has provided further evidence to guide the regulatory Agency in considering more flexible criteria and formulation design space to biowaiver recommendation for non-Q1 same/Q2 similar formulations for putative BCS Class III drug products. Consequently, biowaiver for in vivo BE may be considered for non-Q1 same/Q2 similar formulation for generic drug approval.

Meanwhile, the flexibility in formulation recommendation for BCS Class III biowaiver may raise potential challenges. BCS Class III drug products contain a wide range of intestinal permeability from nearly zero to less than 85%. From a conservative or cautious perspective, the formulation criteria may introduce the new sub-classes for BCS Class III, the low (Fa < 50%) and moderate (50% ≤ Fa < 85%) permeability. Compared to low permeability, in vivo absorption of BCS Class III drug substance with moderate permeability could potentially behave more like BCS Class I drugs. Demonstration of high solubility and very rapid dissolution in multi-media for those generic products would support sufficient similarity to their corresponding reference products. Therefore, BCS Class III drug products with moderate permeability may be considered to extend a biowaiver for non-Q1 same/Q2 similar formulation of an ANDA submission. Generic applicants should provide appropriate documentation regarding moderate permeability via in vitro or in vivo permeable approaches in addition to demonstrating that the product does not contain any excipients that will affect the rate or extent of the drug’s absorption. Whereas, if low permeability drug substance contains non-Q1 same/Q2 similar formulation, in vitro dissolution testing only may not be sufficient to assess the impact of different excipients and excipient levels on intestinal absorption in human. In these cases, in vivo fasting and fed BE studies may still be recommended in human subjects. The Agency may remain stringent criteria for formulation similarity, excipients, and different excipient classes on BCS Class III drug with low permeability (Fa < 50%).

## 5. Conclusions

Our results of in vivo BE with PK endpoints would infer that the commonly used excipients collected from 133 approved ANDAs may not affect intestinal absorption of 16 investigated BCS Class III drugs. The rate and extent of absorption of those drugs appears to be more dependent upon the biopharmaceutic and physiologic properties of BCS Class IIII drug substance and less, or not, dependent upon their formulations with different commonly used excipients and excipient classes. The present study may help to explore more flexibility for biowaiver recommendation regarding the conservative formulation criteria of Q1 sameness/Q2 similarity for BCS Class III drug products, particularly for drug substance with moderate permeability. 

## Figures and Tables

**Figure 1 pharmaceutics-15-02366-f001:**
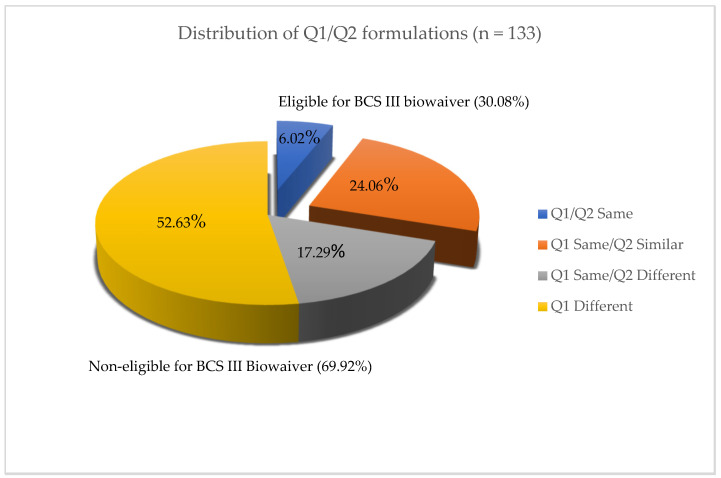
Distribution of diverse Q1 and Q2 formulations for in vivo BE studies. Approximately 30.08% of generic formulations were Q1/Q2 same or Q1 same/Q2 similar and 69.92% of generic drug formulations were non-Q1/Q2 similar.

**Figure 2 pharmaceutics-15-02366-f002:**
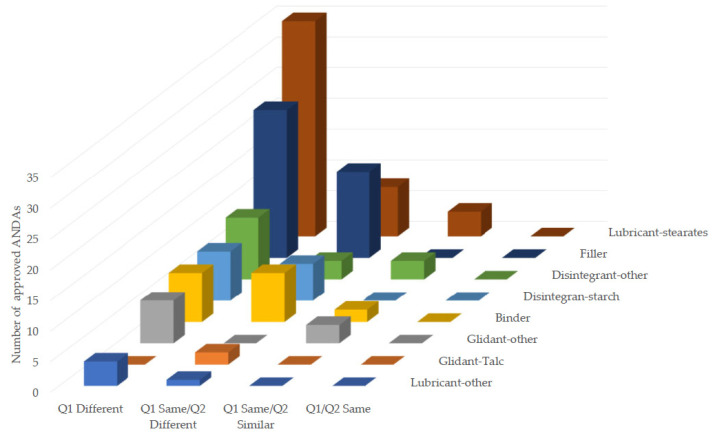
Number of approved ANDAs exceeding BCS Class III-based biowaiver criteria on weight percentile changes (%*w*/*w*) with regard to different excipient class in four formulation groups: Q1 Different, Q1 Same/Q2 Different, Q1 Same/Q2 Similar, and Q1/Q2 Same.

**Figure 3 pharmaceutics-15-02366-f003:**
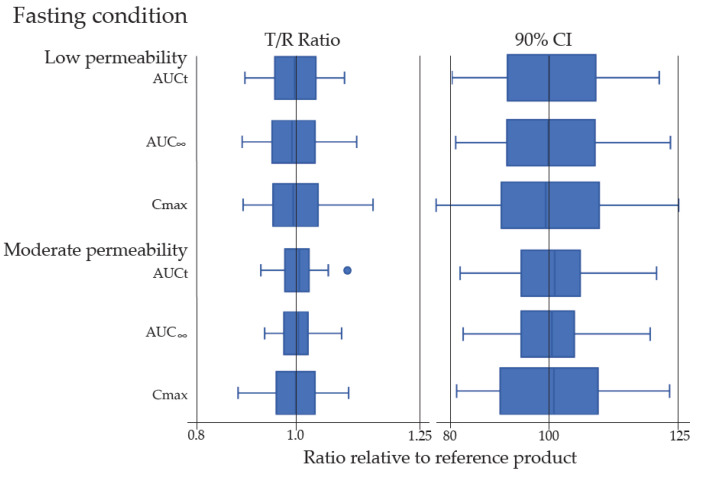
Summary statistics of 217 BE studies under fasting and fed conditions using Box-and whisker plot for the distribution of the T/R ratios and 90% CIs of the T/R geometric mean ratios for C_max_, AUC_t_, and AUC_inf_ by low or moderate permeability.

**Table 1 pharmaceutics-15-02366-t001:** General information of classification and special status for 16 putative BCS Class III drug candidates.

Classification	No. of Drug Substances (Products)	Drug Name	Special Status		
			Orphan	Shortage	Patients
Anti-infection	5 (7)	Oseltamivir, Abacavir, Acyclovir, TDF *, HCQ *	1	3	0
Anti-cardiac	2 (2)	Nadolol, Atenolol,	0	2	0
Chelator	2 (3)	Penicillamine, Trientine	1	1	0
Other **	7 (7)	Colchicine, Rasagiline Ranitidine, Tipiracil HCl, Trifluridine, Metformin Migalastat	2	0	1

* TDF: Tenofovir disoproxil fumarate; HCQ: Hydroxychloroquine sulfate. ** Other including antigout, anti-parkinsonian, H2 blocker, antidiabetic, anti-Fabry disease, and anti-oncology drugs (Data source: Drugs@FDA: FDA-Approved Drugs).

**Table 2 pharmaceutics-15-02366-t002:** Solubility data for 16 putative BCS Class III drug substance in different pH media.

Drug Substance	pH Range	Lowest Solubility Value (mg/250 mL)	Highest Single Dose (mg)
Abacavir	1.2 to 8	23,436	600
Acyclovir	1.2 to 7.5	550 *	800
Atenolol	1.2 to 7.5	5075.75	200
Colchicine	1.2 to 6.8	3337.5	2.4
HCQ	1.2 to 7.4	105,300	600
Metformin	1.2 to 7.6	95,812.5	2000
Migalastat	1.2 to 7.5	31,404.25	123
Nadolol	1.6 to 6.9	487	240
Oseltamivir	1.2 to 7.5	43,252.5	75
Penicillamine	1.2 to 6.8	25,660	1000
Ranitidine	1.2 to 6.8	180,000	300
Rasagiline	1 to 6.8	58,945	1
TDF	1.2 to 7.5	1617.5	300
Tipiracil **	1.0 to 7.5	30,000	37.68
Trifluridine **	1.0 to 7.5	15,000	80
Trientine	1.2 to 7.5	1250	1000

* This drug may not be highly soluble. However, based on published studies, it may be considered as highly soluble. ** This is a fixed dose combination dosage form.

**Table 3 pharmaceutics-15-02366-t003:** The information of permeability and efflux transporters for 16 putative BCS class III drug substance candidates.

Permeability	No. of Drug Substances	Drug Substance	Absorption	Permeability Rate	Substrate(s) of Efflux Transporter(s)	Permeability Determination
Moderate (fa ^1^ = 50–84%)	8	Metformin	Rapid	50–60%	OCT ^2^	Absolute BA
		Atenolol	Rapid and Consistent	50%	P-gp ^3^, OATP ^4^	Absolute BA
		Ranitidine	Rapid	50%	P-gp, OCT	Absolute BA
		Abacavir	Rapid	83%	P-gp, BCRP ^5^	Absolute BA
		Oseltamivir	Rapid	80%	P-gp	Absolute BA
		HCQ	Slow	67–74%	N/A	Absolute BA
		Migalastat	Rapid	75%	SGLT1 ^6^	Absolute BA
		Trifluridine	Rapid	55%	N/A	Mass balance
Low (fa < 50%)	8	Rasagiline	Rapid	36%	N/A	Absolute BA
		Nadolol	Slow	30%	P-gp	Absolute BA
		TDF	Rapid	25%	P-gp, BCRP	Absolute BA
		Acyclovir	Slow	10–20%	P-gp, OCT	Absolute BA
		Penicillamine	Rapid	40–70%	OATP	Absolute BA
		Colchicine	Rapid	45%	P-gp	Absolute BA
		Tipiracil	Rapid	27%	N/A	Mass balance
		Trientine	Slow	6–18%	N/A	Mass balance

^1^ fa: in vivo extent of absorption; ^2^ OCT: Organic cation transporters; ^3^ P-gp: P-glycoprotein; ^4^ OATP: Organic anion transporting polypeptide; ^5^ BCRP: Breast cancer resistance protein; ^6^ SGLT1: Sodium-glucose cotransporter 1 (Data source: Drugs@FDA: FDA-Approved Drugs).

**Table 4 pharmaceutics-15-02366-t004:** Excipients listed at least 5% of approved ANDAs under investigation collected from 133 formulations.

Function	Excipient	Total of ANDAs	% of Total	Range of Weight per Unit (mg)	Range (%*w*/*w*)
Filler	Microcrystalline Cellulose	72	54.14	13.96–344.85	1.83–74.97
	Lactose	44	33.08	25–277	11.0–69.02
Disintegrant	Starch	35	26.32	10–188	2.39–42.5
	Croscarmellose Sodium	31	23.31	2.0–80	1.04–10.0
	Sodium Starch Glycolate	44	33..08	3.0–74.2	0.99–10.0
	Crospovidone	10	7.52	2.11–65.0	0.2–16.41
Binder	Povidone	45	33.84	1.25–90	0.27–7.75
	Pregelatinized starch	38	28.57	4–136.4	1.0–57.02
	Hypromellose	7	5.26	1.0–25.0	0.5–5.8
Lubricant	Magnesium Stearate	103	77.44	0.5–20	0.25–3.33
	Sodium Stearyl Fumarate	8	6.02	0.6–18.0	0.24–3.0
	Stearic acid	22	16.54	1.0–6.0	0.5–16.67
Surfactant	Sodium lauryl sulfate	18	13.53	0.31–6.3	0.1–1.5
Glidant	Colloidal Silicon Dioxide	44	33.08	0.4–15.6	0.16–2.52
	Talc	17	12.78	1.25–10	0.5–5.03
Acidifier	Citric acid	10	7.52	1.0–21.88	0.28–4.76

Total number of ANDAs = 133.

**Table 5 pharmaceutics-15-02366-t005:** Unusual excipients that may have potential impact on absorption.

Function	Excipient	Total of ANDAS	% of Total	Range of Weight per Unit (mg)	Range (%*w*/*w*)
Binder	HPMC	7	5.34	1.0–25.0	0.5–5.8
Surfactant	SLS	18	13.74	0.31–6.3	0.1–1.5
	Polysorbate 80	2	1.53	3.0–3.0	0.45–1.0
Sugar alcohol	Isomalt	1	0.76	114.24	95.2
	Mannitol	4	3.05	44.0–111.85	6.29–48.53
Osmotic agent	PEG 3350	1	0.76	1.0	0.09

Total number of ANDAs = 133.

**Table 6 pharmaceutics-15-02366-t006:** Criteria for weight percentile changes (%*w*/*w*) in different excipient class for drug products containing BCS Class III drugs.

Excipient Class	% Difference Relative to Core Weight * (*w*/*w*)
Filler	10%
Disintegrant-starch	6%
Disintegrant-other	2%
Binder	1%
Lubricant-stearates	0.5%
Lubricant-other	2%
Glidant-Talc	2%
Glidant-other	0.2%

* Note: Core does not include tablet film coat or capsule shell. (Data source: Guidance for Industry on M9 BCS-Based Biowaiver)

**Table 7 pharmaceutics-15-02366-t007:** Summary of 217 fasting and fed BE studies from 133 approved ANDA submissions.

Permeability	No. of Drug Product	No. of ANDA	BE Study Design	Subject	Fasting	Fed	Total
Moderate	8 (tablet and capsule)	54	Two-way crossover	Healthy	54	38	92
Low	11 (tablet and capsule)	79	Two-way crossover/Three-way replicated	Healthy/Patient *	79	46	125
Total	19	133			133	84	217

* One ANDA contains multi-dose steady state BE study in patients for tipiracil hydrochloride; trifluridine tablets.

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
