# Peer review of "Effect of the Similarity of Formulations and Excipients of Approved Generic Drug Products on In Vivo Bioequivalence for Putative Biopharmaceutics Classification System Class III Drugs"

_pharmaceutics, 2023, doi:10.3390/pharmaceutics15092366_

Round 1
Reviewer 1 Report
the manuscript is coherent and easy to read with respect to all the information it provides. There are several aspects of form that I believe would make the manuscript easier to read and analyze:
1. Figure 2 could be subdivided into 4 figures, for each group of different Q1/Q2 formulations.
2. On page 8, the figure should be number 4 and not number 1.
3. Figure 3 could improve the image quality.
4. Due to the significance of this work, it would be very useful if the information provided in the tables were associated with the references from which the information was obtained.
Author Response
Please find your comments and our responses shown below. Also please refer to our revised version of manuscript by referring the attachment.
Comments from Reviewer 1:
- Figure 2 could be subdivided into 4 figures, for each group of different Q1/q2 formulations:
Response from Authors: We appreciate your suggestion. We have redesigned with some changes made for Figure 2 accordingly in order to show data much clearer. Please refer the revised Figure 2 in the content of manuscript.
- On page 8, the figure should be number 4 and not number 1.
Response from Authors: Since figure and table are two separate categories with their corresponding numbering system, we align the number of figure and table separately. Thus, on page 8, this is the first figure for this manuscript. It should be Figure 1.
- Figure 3 could improve the image quality
Response from Authors: We appreciate your suggestion. We have made the changes for Figure 3 accordingly. Please refer the revised Figure 3 in the content of manuscript.
- Due to the significance of this work, it would be very useful if the information provided in the table were associated with the reference from which the information was obtained
Response from Authors: We appreciate your suggestion. We add website link for table 1, 3, and 6 with the reference from which the information was obtained. The data of Table 2, 4, 5, and 7 were obtained from our collected data from abbreviated new drug applications (ANDAs) and new drug applications (NDAs).

Reviewer 2 Report
This article describes a very informative and interesting topic with many researchers in academics and industries that will benefit. The authors have planned and well designed the work and made enough analysis to have a specific conclusion. I appreciate the authors for their efforts to provide the update of the revised publication of the M9 BCS-Based Biowaiver Guidance for Industry. I have a few minor comments the authors can include to improve this manuscript.
Comments
1. The authors considered 16 reputed BCS III class drugs with 19 drug products via 133 approved abbreviated new drug applications in the last decade. Authors can mention whether this selection has omitted any other BCS class III drugs or whether these are the only available drugs. Also, describe if there are any other specific criteria the authors considered while selecting these 133 approved abbreviated new drug applications.
2. Line 107, Furthermore, in vivo BE approaches may recruitment of a large number…. Is not clear. Please revise.
3. Figure 3, is it possible to include statistical significance?
4. Minor typo errors needs correction.
Author Response
Please find your comments and our responses shown below. Also please refer to our revised version of manuscript by referring the attachment.
Comments from Reviewer 2:
- The author considered 16 reputed BCS Class III drugs with 19 drug products via 133 approved ANDAs in the last decade. Author can mention whether this selection has omitted any other BCS Class III drugs or whether these are the only available drugs. Also describe if there are any other specific criteria the authors considered while selecting these 133 approved ANDAs.
Response from Authors: We appreciate your suggestion.
The BCS Class III drugs were selected based on the published literatures, WHO BCS Class I-IV list, FDA internal assessments (the list of BCS Class III by FDA BCS Class Committee), and physicochemical properties (high solubility and low permeability) of specific drug substance from the RLD labeling. We tried to select all BCS Class III drugs from approved ANDAs during the time period from 2006 to 2022.
The drugs that cannot be delivered to the systemic circulation with almost zero permeability (such as GI locally active drugs), narrow therapeutic index drugs, and drugs with unstable solubility & permeability with >10% degradation were excluded from our investigated list.
Please refer to the Abstract on Page 1, we have elaborated the above rationale.
- Line 107, Furthermore, in vivo BE approaches may recruitment of a large number……. Is not clear. Please revise.
Response from Authors: Thanks for your suggestion, we revised it to “Furthermore, in vivo BE with PK endpoint or CCEB approaches may need to recruit a large number of human subjects, which could significantly increase the financial investment burden and unnecessary human exposure.”

Reviewer 3 Report
The authors presented a manuscript entitled “Effect of the Similarity of Formulations and Excipients of Approved Generic Drug Products on In Vivo Bioequivalence for Putative Biopharmaceutics Classification System Class III Drugs” in which many generic and reference medication bioequivalence tests are compared, within 4 groups that have variations related to the qualitative and quantitative excipient similarity between reference/generic. The idea of the article is interesting, and the scientific design was sound. The English is good, but can be improved to increase clarity within the text. The text itself, to my point of view, needs to be clearer; the focal point of the study which is the division of the four groups relating to qualitative and quantitative excipient similarity is not well defined. Later on, the information regarding the comparison of the groups and the correlation between generic reference, is not much clear as well. The figures have bad resolution for the employed fonts. There is not a clear separation between methods in methodology, along with statistical analysis.
Abstract:
ANDA is undefined
L21 “40 different compendial excipients without atypical amount defined by our own data-driven criteria” In don’t quite understood the meaning of this phrase, it is not clear, mostly the ‘’without atypical amount’’
L23 “the investigated generic formulations met Q1 the same” Again highly unclear statement
L27 “excipients without atypical amounts did not impact absorption of 16 putative BCS Class III drugs.” Without atypical would mean typical, correct? Typical related to what?
L131 “biopharmaceutic and physiologic properties” Unclear
L141-146 What to the authors consider “same” and “similar”? What is the threshold, please add a brief and precise explanation of the terms
L161 The authors could also perform an ANOVA and t-Student to compare the statistical difference between generic and reference samples
Resolution in Figure 3 was quite bad for the font size, I couldn’t quite understand what data was related to the generic samples and which was related to the reference
Is there a figure relating the comparison between all 4 proposed groups?
The English is good, but can be improved to increase clarity within the text.
Author Response
Please find your comments and our responses shown below. Also please refer to our revised version of manuscript by referring the attachment.
Comments from Reviewer 3:
- L21 “40 different compendial excipients without atypical amount defined by our own data-driven criteria. In don’t quite understood the meaning of this phrase, it is not clear mostly the “ without atypical amount”.
Response from Authors: We revised it to “Based upon all 133 approved generic formulations in this study, the highest amount of each different compendial excipient with a total of 40, is defined as its corresponding typical amount which does not have potential impact on in vivo drug absorption.”
- L23 “the investigated generic formulations met Q1 the same” Again highly unclear statement
Response from Authors: Based on M9 guidance for industry on BCS system-based biowaivers (2021), qualitative (Q1) the same means that the generic drug uses identical or the same excipients in its formulation as these of the reference.
- L27 “excipients without atypical amounts did not impact absorption of 16 putative BCS Class III drugs. Without atypical amounts mean typical, correct? Typical related to what?
Response from Authors: Yes, without atypical amounts mean typical. In other words, the amount of excipient was not over the highest amount of excipients in 133 approved ANDA formulations. Based on the acceptable bioequivalence results, typical amount of excipients may not impact on in vivo absorption of 16 putative BCS Class III drugs.
- L131 “the biopharmaceutic and physiologic properties” Unclear
Response from Authors: We revised it to “the biopharmaceutic and physiochemical properties”.
- L141-146 What the authors consider “same” and “similar”? What is the threshold, please add a brief and precise explanation of the teams.
Response from Authors: Based on M9 guidance for industry on BCS system-based biowaivers (2021), qualitative (Q1) the same means that the generic drug uses identical or the same excipients in its formulation as the reference. Quantitatively same means individual excipient difference within ±5% [each excipient (%) = (1-T/R)*100 ≤±5% ]. Quantitatively similar means within ± 10% of the amount of excipient in the reference product. Additionally, the cumulative difference for excipients that may affect absorption should be within ± 10%. We added a brief summary in the content of manuscript.
- The authors could also perform an ANOVA and t-Student to compare the statistical different between generic and reference samples.
Response from Authors: Generally, we used ANOVA to compare the statistical difference of AUC and Cmax between generic and reference drug products in in vivo BE studies, such as 3.5 In vivo bioequivalence studies in the Section of Results.
- Resolution in Figure 3 was quite bad for the font size, I couldn’t quite understand what data was related to the generic samples and which was related to the reference.
Is there a figure relating the comparison between all 4 proposed groups?
Response from Authors: Thanks for your suggestion, we made the changes for Figure 3 accordingly with a much higher resolution as well as front size. Please refer the revised Figure 3 in the content of manuscript. This figure mainly showed the distribution of 90% confidence interval and T/R ratios of AUCt, AUCinf, and Cmax within BE acceptable criteria ( 80-125% for 90% CI and 0.8-1.25 for T/R ratios) by ANOVA statistical method under fasting and fed conditions, where comparisons are between the generic and reference products.
Figure 1 and Figure 2 may be some comparison among all 4 proposed formulation groups: Q1 Different, Q1 Same/Q2 Different, Q1 Same/Q2 Similar, and Q1/Q2 Same.

Round 2
Reviewer 3 Report
The authors attended to each raised suggestion properly.
English is fine.